

# A chimeric vector for dual use in cyanobacteria and *Escherichia coli*, tested with cystatin, a nonfluorescent reporter protein

Mojca Juteršek[1,2] and Marko Dolinar[1]

[1] Faculty of Chemistry and Chemical Technology, University of Ljubljana, Ljubljana, Slovenia
[2] Current Affiliation: National Institute of Biology, Ljubljana, Slovenia

## ABSTRACT

**Background:** Developing sustainable autotrophic cell factories depends heavily on the availability of robust and well-characterized biological parts. For cyanobacteria, these still lag behind the more advanced *E. coli* toolkit. In the course of previous protein expression experiments with cyanobacteria, we encountered inconveniences in working with currently available RSF1010-based shuttle plasmids, particularly due to their low biosafety and low yields of recombinant proteins. We also recognized some drawbacks of the commonly used fluorescent reporters, as quantification can be affected by the intrinsic fluorescence of cyanobacteria. To overcome these drawbacks, we envisioned a new chimeric vector and an alternative reporter that could be used in cyanobacterial synthetic biology and tested them in the model cyanobacterium *Synechocystis* sp. PCC 6803.

**Methods:** We designed the pMJc01 shuttle plasmid based on the broad host range RSFmob-I replicon. Standard cloning techniques were used for vector construction following the RFC10 synthetic biology standard. The behavior of pMJC01 was tested with selected regulatory elements in *E. coli* and *Synechocystis* sp. PCC 6803 for the biosynthesis of the established GFP reporter and of a new reporter protein, cystatin. Cystatin activity was assayed using papain as a cognate target.

**Results:** With the new vector we observed a significantly higher GFP expression in *E. coli* and *Synechocystis* sp. PCC 6803 compared to the commonly used RSF1010-based pPMQAK1. Cystatin, a cysteine protease inhibitor, was successfully expressed with the new vector in both *E. coli* and *Synechocystis* sp. PCC 6803. Its expression levels allowed quantification comparable to the standardly used fluorescent reporter GFPmut3b. An important advantage of the new vector is its improved biosafety due to the absence of plasmid regions encoding conjugative transfer components.

The broadhost range vector pMJc01 could find application in synthetic biology and biotechnology of cyanobacteria due to its relatively small size, stability and ease of use. In addition, cystatin could be a useful reporter in all cell systems that do not contain papain-type proteases and inhibitors, such as cyanobacteria, and provides an alternative to fluorescent reporters or complements them.

Corresponding author
Marko Dolinar,
marko.dolinar@fkkt.uni-lj.si

## INTRODUCTION

Cyanobacteria are emerging as versatile hosts for synthetic biology. Starting as poorly adapted microorganisms for genetic engineering, advances in the last decade have made them attractive organisms for biosynthetic production of commodities ranging from biofuels to high added-value compounds. A book (*Zhang & Song, 2018*), several book chapters (*e.g.*, *Klemenčič et al., 2017*; *Badary & Sode, 2020*; *Hudson, 2021*) and review articles (*Khan et al., 2019*; *Lindblad et al., 2019*; *Santos-Merino, Singh & Ducat, 2019*; *Vavitsas et al., 2019*; *Ng, Keskin & Tan, 2020*; *Wang, Gao & Yang, 2020*; *Liu et al., 2021*; *Sproles et al., 2021*) published in recent years emphasize the potential of cyanobacteria and present encouraging results obtained with these hosts.

To realize their full application potential as green engineered biofactories, a set of well-characterized genetic elements is necessary to provide a starting material for higher-order synthetic systems. Although the arsenal of available and characterized regulatory sequences for cyanobacteria is expanding, there is still an urgent need to develop stronger and better regulated promoters, ribosome binding sites and terminators. For this, we need reliable reporter systems that allow quantification of the effects achieved by novel genetic elements, and a reliable plasmid backbone.

Fluorescent proteins are most commonly used as reporters in cyanobacteria because they allow both high-throughput analyses and *in vivo* measurements. Despite their clear advantages and popularity, some properties make them less suitable for certain experimental setups. Fluorophore formation is oxygen-dependent and reporter activity can therefore vary with oxygen levels in the cell. In addition, fluorescent proteins can have problems with sensitivity due to the lack of signal amplification and background autofluorescence. This is particularly critical in cyanobacteria with strong intrinsic fluorescence from photosynthetic pigments, which can also act as fluorescence quenchers (*Taniuchi, Murakami & Ohki, 2008*). An alternative reporter system based on a different measurement technique could counteract the disadvantages of fluorescence reporters and improve the reliability of the measurements, thus contributing to a better characterization of biological parts.

Essential to modern cyanobacterial biotechnology is the introduction of new genetic elements into host strains. There are two general approaches to introduce such elements into cyanobacteria: genomic insertion and episomal introduction. The former is complicated by a limited number of true neutral sites in the genome (*Pinto et al., 2015*) and by a low number of genome copies. Where multiple copies of the genome exist, as in the best-studied unicellular cyanobacterium *Synechocystis* sp. PCC 6803 (*Zerulla, Ludt & Soppa, 2016*), obtaining strains in which all genomic copies are genetically modified is time-consuming and requires quantitative analyses (*Hitchcock, Hunter & Canniffe, 2020*). Therefore, episomal expression is often preferred, especially for characterization of biological parts and protein expression. Although many vectors have been developed, they are not generally suitable to every experimental setup.

The first group of cyanobacterial vectors is based on replicators from endogenous cyanobacterial plasmids that lack broad host range fitness. This can be overcome by using

RSF1010-based plasmids, but their biological safety is questionable (*Wright, Stan & Ellis, 2013*), as they allow conjugative transfer; mobilization is possible by a variety of different type IV transporters (*Meyer, 2009*). In addition, RSF1010-derived plasmids have low copy number in both *E. coli* and cyanobacteria. In *E. coli*, copy numbers are probably five or even less (*Jahn et al., 2016*), although 10–12 copies per chromosome copy were originally reported (*Frey & Bagdasarian, 1989*). Similarly low copy numbers have been reported in cyanobacteria: 10 copies per cell in *Synechocystis* sp. PCC 6803 and PCC 6714 and *Synechococcus* elongatus PCC 7942 (*Mermet-Bouvier et al., 1993*) or one plasmid copy per genome copy for pPMQAK1 vector in *Synechocystis* sp. PCC 6803 (*Huang et al., 2010*; *Jin et al., 2018*).

Low plasmid copy number could be reflected in lower protein expression levels compared to high copy number plasmids. The correlation between plasmid copy number and reporter expression was clearly demonstrated by *Jahn et al. (2016)* for RSF1010-based plasmids in *E. coli*, where two subpopulations of cells with different levels of EGFP expression were detected. A similar observation was reported by *Thompson et al. (2018)* on pSC101 ori mutants. Such correlation has also been observed in other bacteria, *e.g.,* *Corynebacterium glutamicum* (*Hashiro & Yasueda, 2018*), while quantitative data for cyanobacterial hosts are still lacking. A difference in vector copy number was suspected for differences in reporter expression in *Anabaena* sp. PCC 7120 (*Ma, Schmidt & Golden, 2014*), while in *Synechococcus elongatus* PCC 7942, reporter expression level correlated well with vector copy number (*Chen et al., 2016*).

Conjugation is a process in which bacteria exchange their genetic material and in which plasmids act as vectors. Although conjugation is commonly used to introduce plasmid vectors into cyanobacterial cells, usually by exploiting the so-called triparental mating (*Vioque, 2007*), cyanobacteria can also be transformed by electroporation and some strains are even naturally competent for transformation (*Zang et al., 2007*). In our experiments with the model cyanobacterium *Synechocystis* sp. PCC 6803, we use electroporation exclusively. Therefore, we could easily trade off the conjugation-related genetic elements on the vector for improved biosafety and smaller size.

Our goal was to (a) construct a shuttle vector with increased copy number and improved biological safety and (b) present a reporter that is independent of intrinsic fluorescence or other activity in cyanobacteria. For the vector, we chose the pPMQAK1 plasmid as a starting point for modifications. This vector is a commonly used RSF1010-based shuttle plasmid for cyanobacterial synthetic biology and has been used in our lab for several years. However, to improve its biological safety, we wanted to retain regions that are required for replication in different bacterial hosts, but not those that enable conjugative transfer (oriT and *mob* regions). Therefore, we decided to complement selected pPMQAK1 regions with those of RSFmob-I, a non-mobilizable RSF1010 derivative developed by *Katashkina et al. (2007)*. We inserted into the new vector the kanamycin resistance gene and a standardized synthetic biology cloning site compatible with RFC[10] assembly (*Knight, 2003*; *Shetty, Endy & Knight, 2008*).

The constructed pMJc01 vector backbone was tested and characterized in both *E. coli* and *Synechocystis* sp. PCC 6803. First, the common GFPmut3b fluorescent reporter

(*Cormack, Valdivia & Falkow, 1996*) was used to demonstrate the usability of the vector. Then, we used the improved vector backbone to investigate the conformity of the vector with cystatin as a new non-fluorogenic reporter. Cystatin is a stable protein first described as a hen egg-white protease inhibitor (*Fossum & Whitaker, 1968*). We chose cystatin (reviewed in *Turk, Stoka & Turk, 2008*) as a potential reporter because it can be easily assayed with an enzyme inhibition test using papain (a cysteine protease) as the target enzyme. The enzyme-based detection method also allows for signal amplification and thus higher sensitivity. We found cystatin to be a useful reporter both in *E. coli* and in *Synechocystis* sp. PCC 6803. It could be used as an alternative or to complement fluorescent protein reporters.

## MATERIALS AND METHODS

### Strains and growth conditions

Cultures of the cyanobacterium *Synechocystis* sp. PCC 6803 (*S.* sp. PCC 6803, obtained from Pasteur Culture Collection of Cyanobacteria) were grown on BG11 agar plates or in BG11 liquid medium (*Stanier et al., 1971*) at room temperature under continuous white light of 10–30 µmol photons $m^{-2}$ $s^{-1}$. Liquid cultures were shaken at 100 rpm. For engineered strains transformed with pMJc01 or pPMQAK1 plasmid or their derivatives, the growth medium was supplemented with 50 µg/ml kanamycin.

*Escherichia coli* XL1-Blue was used for cloning and characterization of vector constructs. All strains were cultured in liquid LB medium with shaking at 140 rpm at 37 °C or on LB agar plates. Cells transformed with pMJc01, pSB3K3 or pPMQAK1 were grown in a medium containing 50 µg/ml kanamycin and those transformed with the RSFmob-I plasmid were cultured in the presence of 50 µg/ml streptomycin.

For the transformation of *E. coli*, we used chemically competent cells and for the transformation of *S.* sp. PCC 6803, we used electroporation. In brief, cells in the exponential growth phase were washed three times in ice-cold sterile Milli-Q water (followed by centrifugation for 10 min at 4 °C and 6,000*g*) and mixed with 0.5–2 µg of plasmid dissolved in 40 µl of sterile Milli-Q water. The suspension of cells and plasmid was transferred to electroporation cuvettes with a gap width of two mm and subjected to an electrical pulse of 2,500 V for five ms (with the capacitance set to 25 µF and the resistance set to 200 Ω). After pulse treatment, cells were immediately resuspended in five ml of BG11 medium without kanamycin and left to recover overnight at lower light intensity. The next day, the liquid cultures were poured onto BG11 agar plates supplemented with five µg/ml kanamycin. The resulting colonies were transferred to BG11 agar plates supplemented with gradually increasing concentrations of kanamycin (10, 25 and finally 50 µg/ml).

### Construction of the broad host range shuttle vector pMJc01

A DNA region comprising sequences required for autonomous self-replication was PCR-amplified with primers RSFmobF and RSFmobR (Table 1) from the template plasmid RSFmob-I (*Katashkina et al., 2007*) (available *via* Russian National Collection of Industrial Microorganisms). A DNA fragment containing a synthetic biology RFC[10] standard

**Table 1 List of primers used in this study.** All primers (except VF2 – BBa_G00100 and VR – BBa_G00101, see Registry of Standard Biological Parts, http://parts.igem.org/Main_Page) were designed as part of this study. Underlined are 18 nt long overlapping sequences that were used for the overhang assembly of pMJc01.

| Primer name | Sequence |
| --- | --- |
| RSFmobF | AACTGTCACGAACCCCTGCAATAACTGT |
| RSFmobR | TTTGTTGAATGGGTCAGCCTGCCGCCTT |
| pSB3K3F | GCTGACCCATTCAACAAAGCCACGTTGTGT |
| pSB3K3R | TTCCATGGTGCCACCTGACGTCTAAGA |
| L21_RBS*_F | GCTCTAGAGGGGCCCTCCCTATCAGTGATAGAGATTGACATCCCTATC AGTGATAGATATAATGGGAGCTACTAGAGTAGTGGAGGTTACTAGTCG |
| L21_RBS*_R | CGACTAGTAACCTCCACTACTCTAGTAGCTCCCATTATATCTATCACT ATAGGGATGTCAATCTCTATCACTGATAGGGAGGGCCCCTCTAGAGC |
| GFPmut3bF | GCTCTAGATGCGTAAAGGAGAAGA |
| VR | ATTACCGCCTTTGAGTGAGC |
| VF2 | TGCCACCTGACGTCTAAGAA |

cloning site and a kanamycin resistance cassette was PCR-amplified from the pSB3K3 template plasmid using primers pSB3K3F and pSB3K3R (Table 1). Primers RSFmobR and pSB3K3F contained an 18 bp long complementary sequence that allowed overlap extension PCR for splicing of the 3,771-bp fragment from plasmid RSFmob-I and the 2,238-bp fragment from plasmid pSB3K3 to plasmid pMJc01. The corresponding linear 5,999-bp product was phosphorylated at the 5′ ends, circularized with T4 DNA ligase, and used to transform competent *E. coli* cells. The sequence of the new pMJc01 vector was verified by Sanger sequencing (GATC Biotech, Eurofins Genomics).

## Construction of GFPmut3b or cystatin reporter-expressing plasmids based on pMJc01, pPMQAK1 and pSB3K3

For comparative characterization of the new vector and evaluation of cystatin as a reporter, we constructed plasmids pMJc01, pPMQAK1, and pSB3K3 without reporter-expressing sequences (EV) and with 3 different expression cassettes (Table 2). For the first cassette, we chose BBa_I20260 (all BBa codes refer to the Registry of Standard Biological Parts, http://parts.igem.org/Main_Page, from which they can be retrieved as sequences or as physical DNA inserted into a plasmid backbone), a reference standard construct for relative promoter activity measurements based on GFPmut3b (*Kelly et al., 2009*). Plasmids pMJc01 and pSB3K3 lacking BBa_I20260 were obtained by *Xba*I/*Spe*I restriction followed by ligation. For plasmid pPMQAK1, we started from the pPMQAK1_EV plasmid (BBa_J153000), which was previously prepared in our laboratory. To make the pPMQAK1_BBa_I20260 plasmid, we cut the pPMQAK1_EV backbone with *Eco*RI. and *Pst*I. The same restriction endonucleases were used to excise BBa_I20260 from pMJc01 and clone it into the pPMQAK1 plasmid backbone.

Since the regulatory sequences in BBa_I20260 (BBa_J23101 promoter and BBa_B0032 RBS) show low activity in *Synechocystis* sp. PCC 6803 (*Huang & Lindblad, 2013*; *Englund, Liang & Lindberg, 2016*), we additionally prepared reporter constructs under the

**Table 2 List of constructed plasmids.** Each plasmid backbone was constructed as an empty vector (EV) as well as with three different expression cassettes. The last two columns indicate whether the constructs were successfully transferred to and tested in either microorganism.

| Plasmid backbone | Regulatory element | Reporter protein | E. coli | S. sp. PCC 6803 |
|---|---|---|---|---|
| **pMJc01** | / | / | yes | yes |
| | J23101_B0032 | GFPmut3b | yes | yes |
| | L21_RBS* | GFPmut3b | yes | yes |
| | L21_RBS* | cystatin | yes | yes |
| **pSB3K3** | / | / | yes | yes |
| | J23101_B0032 | GFPmut3b | yes | yes |
| | L21_RBS* | GFPmut3b | yes | yes |
| | L21_RBS* | cystatin | yes | yes |
| **pPMQAK1** | / | / | yes | yes |
| | J23101_B0032 | GFPmut3b | yes | no |
| | L21_RBS* | GFPmut3b | yes | no |
| | L21_RBS* | cystatin | yes | yes |

control of the L21_RBS*regulatory element. The designed regulatory element was made from the complementary oligonucleotides L21_RBS*_F and L21_RBS*_R and comprises the −35 element (5′-TTGACA-3′), the −10 element (5′-TATAAT-3′), and two tetO2 operators: one upstream of the −35 element and one between the −35 and −10 elements (Table 1). We chose L21 (*Huang & Lindblad, 2013*) because the expression of the reporter in *Synechocystis* was about 10-fold stronger compared to J23101 and because it did not respond to anhydrotetracyline, which was desirable in our case because we were looking for a constitutive reporter.

The coding sequence of the cystatin reporter was designed based on the amino acid sequence deposited in the UniProt database (chicken cystatin, accession number P01038). We included residues 24–139, which constitute a functional protein with 2 disulfide bridges, and reverse-translated them into the nucleotide sequence. An *Xba*I restriction site with start codon (5′-TCTAG**ATG**-3′) was added to the 5′ end and a hexahistidine tag with stop codon and restriction sites was added to the 3′ end of the cystatin coding sequence (*cystatin6803*). The sequence was codon-optimized and synthesized by Synbio Technologies.

The *GFPmut3b* coding region was PCR-amplified from a recombinant pSB3K3 derivative with primers GFPmut3bF and VR (Table 1) and cut with *Xba*I/*Pst*I. The same restriction endonucleases were used to cut *cystatin6803* inserted by the supplier into the pUC57 plasmid. Complementary oligonucleotides containing the regulatory sequence L21_RBS* were hybridized and cut with *Spe*I. The reporter coding regions and the regulatory fragment were ligated with complementary *Xba*I/*Spe*I overhangs and the product was cut with *Xba*I. The obtained reporter gene expressing cassettes were inserted into plasmid backbones pSB3K3, pMJc01, and pPMQAK1, which were previously linearized with *Xba*I/*Pst*I restriction endonucleases. *E. coli* cells were transformed with the

constructed plasmids and colonies were verified by colony PCR using primers VF2 and VR after overnight incubation (Table 1).

## Characterization of the pMJc01 vector in *E. coli*

The relative reporter expression from pMJc01, pSB3K3, and pPMQAK1 in *E. coli* was determined using GFPmut3b fluorescence at the stationary growth phase in cell lysates and in liquid cultures (similarly to *Huang et al., 2010*). For measurements in liquid cultures, overnight cultures of cells transformed with each of the three plasmid backbones without insert and with BBa_I20260 or L21_RBS*_GFPmut3b reporter constructs were diluted 8-fold to the final volume of 2 ml in LB medium, supplemented with kanamycin (final $Abs_{600}$ 0.10–0.14). Alternatively, for measurements of cell lysates, cells were pelleted from 2 ml overnight cultures and resuspended in TE (10 mM Tris pH 8.0, 1 mM EDTA) buffer, with volume adjusted to achieve equal cell densities in all cell suspensions. Cells were lysed by sonication for $3 \times 10$ s and 35 µl of the cell lysate was resuspended in two ml TE buffer. Fluorescence emission spectra (505–530 nm) were measured in glass cuvettes with 1 cm light path in Perkin Elmer LS50B fluorimeter with the excitation wavelength of 485 nm. The average of five spectra was recorded for each sample and the fluorescence intensity at the emission maximum (518 nm for cultures and 514.5 nm for cell lysates) was read. The values of fluorescence intensity in the cultures were normalized to cell density ($F/Abs_{600}$) and the average $F/Abs_{600}$ ratio of the biological triplicates of the samples with empty plasmids was subtracted from the $F/Abs_{600}$ ratios of the biological triplicates of the same plasmid with reporter construct and the average was calculated from the differences. For cell lysates, the fluorescence intensity value of cells with empty plasmid was subtracted from the intensity values of the triplicates of plasmids with reporter constructs and the average was calculated from the differences.

Quantification of restriction fragments in the agarose gel was used to evaluate the absolute copy numbers of pMJc01, pPMQAK1, and pSB3K3. This approach to plasmid copy number determination is based on the established densitometric technique described by *Projan, Carleton & Novick (1983)*, but we adapted it for comparison of plasmids with different sizes so that only the fluorescence of a restriction fragment of the same size was quantified in all the plasmids. We isolated pMJc01, pPMQAK1, and pSB3K3 plasmids with the L21_RBS*_GFPmut3b insert from approximately $5 \times 10^9$ *E. coli* cells from overnight cultures and used *Not*I for digestion, resulting in linear plasmid backbones and a 943-bp fragment encoding the reporter. Restriction products were loaded alongside various amounts of DNA standards (100 bp DNA ladder from New England Biolabs and GeneRuler 1 kb DNA ladder from Thermo Scientific, Waltham, MA, United States), separated in a 1.5% agarose gel with ethidium bromide, and visualized under UV light. The gel image was analyzed using GelAnalyzer software, and the 943-bp fragment was quantified using a calibration curve generated with known amounts of 1,000-bp fragments in the ladders used. For pMJc01 and pSB3K3, we also performed restriction with *Bsp*HI, resulting in a 2,128-bp fragment from both plasmids, and used known amounts of the 2,000-bp marker for quantification.

Measurements of GFPmut3b fluorescence at different growth stages were performed on *E. coli* transformed with pMJc01_L21_RBS*_*GFPmut3b* and pMJc01_EV. We took culture samples from the liquid cultures (grown in biological triplicates) at different time points and diluted them to $Abs_{600}$ of 0.1. Fluorescence measurements were performed as described above. Since all samples were diluted to equal cell densities, no normalization of fluorescence values was performed.

## Characterization of cystatin reporter in *E. coli*

To assess the expression of recombinant chicken cystatin in *E. coli*, we used an adapted papain inhibition assay (*Kopitar et al., 1978*). After overnight cultivation, cells were pelleted from 1.5 ml of bacterial culture and suspended in TE buffer, with the volume adjusted to achieve equal cell densities in all samples. Cells were lysed by sonication for $3 \times 10$ s and lysates were centrifuged at $14,000g$ for 2 min. For the inhibition assay, 35 µl of the soluble fraction was used together with 175 µl of 100 mM phosphate buffer (pH 6.0), 105 µl of TE buffer and 35 µl of papain solution (35 µg/ml) in phosphate buffer. Two controls were used: a blank without enzyme to determine background absorbance (used to set zero absorbance on the spectrophotometer) and a positive control without cell lysate to confirm the enzymatic activity of the papain used. After incubation of the samples at 37 °C for 5 min, 35 µl of the substrate BANA (*N*-benzoyl-DL-arginine naphthylamide hydrochloride) was added (6.7 mg/ml, freshly diluted 1:5 in phosphate buffer from DMSO stock solution) and incubated at 37 °C for a further 10 min. The enzyme reaction was terminated by adding 385 µl of a 1:1 mixture of Fast Garnet solution in 4% Brij (0.3 mg/ml) and reagent III (50 mM 4-chloromercuribenzoic acid, 60 mM NaOH, 60 mM EDTA, pH 6). After 5 min at room temperature, the absorbance at 550 nm was measured for all samples and the average values for biological triplicates of cells with cystatin expression vector were subtracted from the average values for samples of cells with empty vector.

## Characterization of pMJc01 vector and chicken cystatin reporter in *Synechocystis* sp. PCC 6803

Three biological replicates of *Synechocystis* sp. PCC 6803 transformed with empty pMJc01 or pMJc01 with two GFP reporter constructs (BBa_I20260_*GFPmut3b* or L21_RBS*_*GFPmut3b*) were grown to exponential phase ($Abs_{730}$ = 0.4–0.8) under standard conditions. Depending on the $Abs_{730}$ values, 1–2 ml of the cultures were centrifuged and the pelleted cells resuspended in 300 µl TE buffer. Cells were lysed by sonication ($4 \times 1$ min) and 52.5 µl of the cell lysate was suspended in two ml BG11. This resulted in an equivalent number of cells as used for fluorescence measurements in cultures, with all *S.* sp. PCC 6803 cultures diluted to $Abs_{730}$ = 0.1. Fluorescence measurements were performed as described for *E. coli*.

Measurements of GFPmut3b fluorescence at different growth stages were performed using *S.* sp. PCC 6803 transformed with pMJc01_L21_RBS*_*GFPmut3b* and pMJc01_EV as for *E. coli*. Cells were diluted to equal cell densities ($Abs_{730}$) at all growth stages.
For the papain inhibition assay, cells were prepared as for the GFP fluorescence measurements in cell lysates and 100 μl of the lysate was used for the assay, which was performed as described above for *E. coli*, except that the reaction mixture contained 40 μl of TE buffer instead of 105 μl.

## Sequence submission

Nucleotide sequence of pMJc01 with inserted codon-optimized cystatin expression cassette and L21_RBS* regulatory element was deposited in the GenBank database under accession number MN201591.

# RESULTS

## Construction of pMJc01, a new shuttle vector for cyanobacteria

A broadhost range vector pMJc01 was successfully constructed using overlap extension PCR of fragments from plasmids pSB3K3 (*Shetty, Endy & Knight, 2008*; cloning site and kanamycin resistance cassette) and RSFmob-I (*Katashkina et al., 2007*; oriV and rep regions), as shown in Fig. 1. The originally constructed vector also contained the BBa_I20260 BioBrick with GFPmut3b reporter derived from pSB3K3 and was subsequently used to prepare pMJc01 empty vector (EV) and two additional pMJc01 reporter vectors with either GFPmut3b or cystatin reporters under the transcriptional control of L21_RBS*. We prepared this additional regulatory sequence because the BBa_J23101 promoter and BBa_B0032 RBS in the BBa_I20260 BioBrick show low activity in *Synechocystis* sp. PCC 6803 (*Huang & Lindblad, 2013*; *Englund, Liang & Lindberg, 2016*). The L21 promoter is based on the TetR-repressible promoter BBa_R0040, which previously showed constitutive expression in *S.* sp. PCC 6803 at approximately 9-fold increase in strength compared to BBa_J23101 (*Huang & Lindblad, 2013*). RBS* contains nucleotides complementary to the anti-Shine-Dalgarno sequence of *S.* sp. PCC 6803 and exhibits higher activity in both *E. coli* and *S.* sp. PCC 6803 compared to BBa_B0032 (*Heidorn et al., 2011*; *Englund, Liang & Lindberg, 2016*). For comparison with pMJc01, we also prepared pSB3K3 and pPMQAK1 with the same three reporter cassettes as in pMJc01. For the full list of vectors used, see Table 2.

Sequencing of pMJc01 revealed two same-sense point mutations in the kanamycin resistance gene and a single-base deletion in the His terminator downstream of the cloning site, compared to the sequence of pSB3K3 (retrieved from Registry of Standard Biological Parts). For the mutation detected in the His terminator, we used the RNAfold tool (*Hofacker, 2003*) to determine the possible effects on the secondary structure and function of the terminator. The deleted nucleotide formed a bulge in the double-stranded stem of the original sequence, which led us to conclude that its absence should not affect the functionality of the region. Comparison of the pMJc01 sequence with the RSFmob-I sequence (GenBank ID: EF467360.1) revealed a two-base substitution in *ori*V (CC to AA, positioned between two iterons) and a single-nucleotide deletion and insertion spaced 6 bp apart in the *rep*C region, causing a short frameshift and substitutions of 3 amino acids (E107D, C108A, H109M). The same two point mutations were observed by Huang and coworkers (*Huang et al., 2010*) in the sequence of the pPMQAK1 plasmid, whose *rep*C

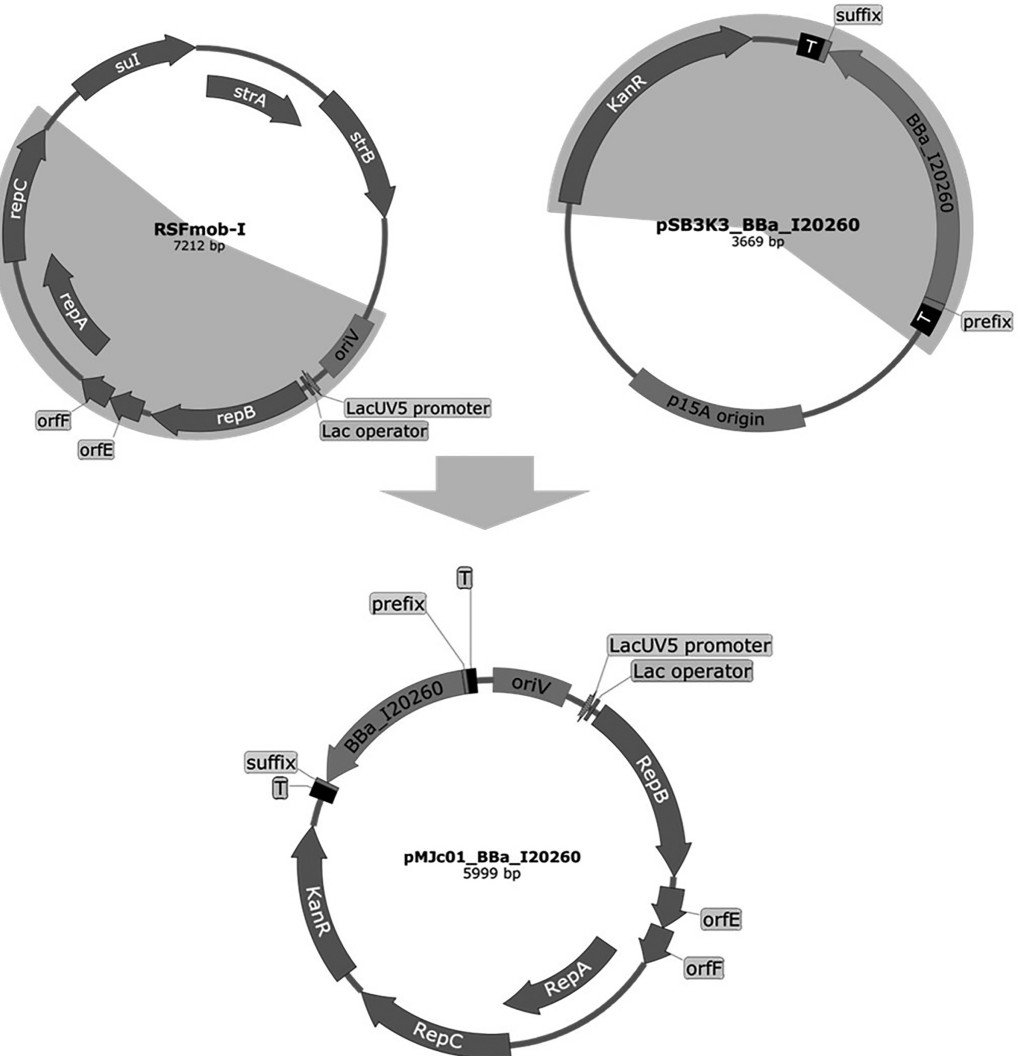

**Figure 1 Plasmid maps of the broad host range pMJc01 vector and its vectors of origin, RSFmob-I and pSB3K3.** The new pMJc01 vector was obtained after splicing by PCR of two fragments amplified from plasmids RSFmob-I and pSB3K3 (marked with grey circular sectors). From RSFmob-I, we amplified the replicator with *ori*V and replicator protein genes (encoding RepB, OrfE, OrfF, RepA, RepC) under the transcriptional control of the *lacUV5* promoter and the *lac* operator. The pSB3K3-derived regions included the kanamycin resistance cassette and the BioBrick cloning site with prefix, suffix, and two terminators (T) at both ends. Our original construct also contained the BBa_I20260 BioBrick. Vector maps were created using SnapGene software (from Insightful Science; available at snapgene.com).

region was derived from pAWG1.1 (another RSF1010 derivative plasmid). This indicates that these are most likely not PCR-generated artefacts, but the exact sequence that was originally incorrectly determined for the RSF1010 plasmid.

## pMJc01 is stable in *E. coli* and allows approximately twofold higher reporter protein expression compared to pPMQAK1

To determine the replication capacity of pMJc01 in *E. coli*, we used the reporter vector pMJc01_L21_RBS*_GFPmut3b (Table 2). We followed GFPmut3b fluorescence at
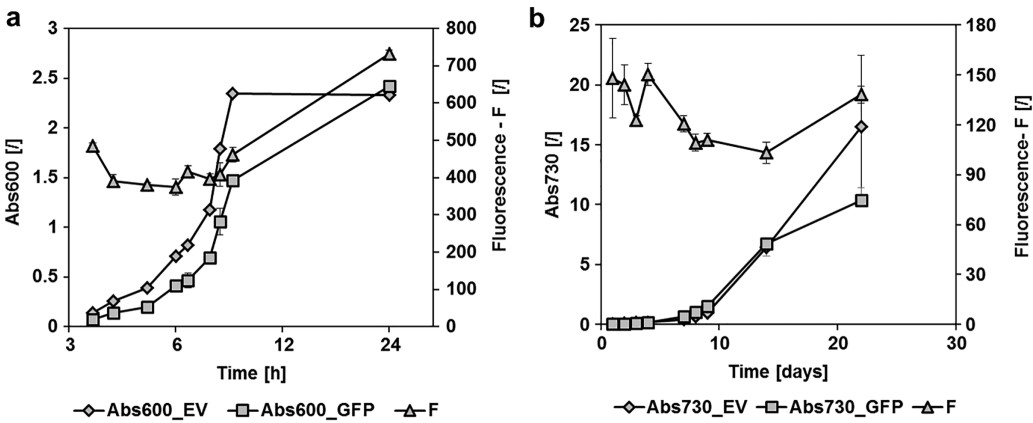

**Figure 2 GFPmut3b fluorescence in cultures of *E. coli* and *Synechocystis* sp. PCC 6803 at different growth stages.** Fluorescence of cultures transformed with empty pMJc01 vector (EV) or pMJc01_L21_RBS*_*GFPmut3b* (GFP) was measured at different time points. Cultures were diluted to Abs600 (for *E. coli*) or Abs730 (for *S.* sp. PCC 6803) of 0.1, except for the first two measurements where absorbance values were lower and the measured fluorescence values were extrapolated to an absorbance of 0.1. Biological triplicates were used (except for *E. coli* transformed with empty pMJc01) and fluorescence values are reported as the difference between the average values for samples with EV and samples with GFPmut3b expression device. Where applicable, standard deviation values are given as error bars. (A) Growth curves of *E. coli* with (GFP, squares) or without (EV, diamonds) GFPmut3b expression device and fluorescence intensity (triangles). The time scale on the *x*-axis is exponential. (B) Growth curves of *Synechocystis* sp. PCC 6803 with (GFP, squares) or without (EV, diamonds) GFPmut3b expression device and fluorescence intensity (triangles).

different time points of cell growth in liquid cultures (Fig. 2A), as described in Methods. Fluorescence was detected at all growth stages with an increase at stationary phase. Over the course of 5 months, we occasionally isolated pMJc01 from *E. coli* and checked its size and integrity by agarose gel electrophoresis. Reporter expression assays were also repeated several times during this period. We observed no changes in the structure of pMJc01 or in the expression levels driven by this plasmid. On the other hand, pMJc01 showed some structural instability when transferred into the *E. coli* DH5α strain, a phenomenon that is detailed and discussed in Supplemental Article S1.

Reporter protein expression in *E. coli* was compared between pMJc01, pPMQAK1, and pSB3K3 by measuring GFPmut3b fluorescence in the stationary growth phase (Fig. 3). We used two GFPmut3b expression cassettes, which allowed validation of the regulatory sequence L21_RBS* and its comparison with the regulatory sequence J23101_B0032 from BBa_I20260, a reference standard construct in promoter strength measurements (*Kelly et al., 2009*).

Results of relative fluorescence intensities for different plasmids were similar when measurements were performed on bacterial cultures or cell lysates. Also, the results were comparable when GFPmut3b was expressed under the control of L21_RBS* or J23101_B0032 regulatory sequences (see *Heidorn et al., 2011* for the description of ribosome binding sites and *Huang & Lindblad, 2013* for promoters). Based on the fluorescence intensities, it can be concluded that pMJc01 allows about twofold higher expression compared to pPMQAK1. The expression from pSB3K3 was slightly higher than

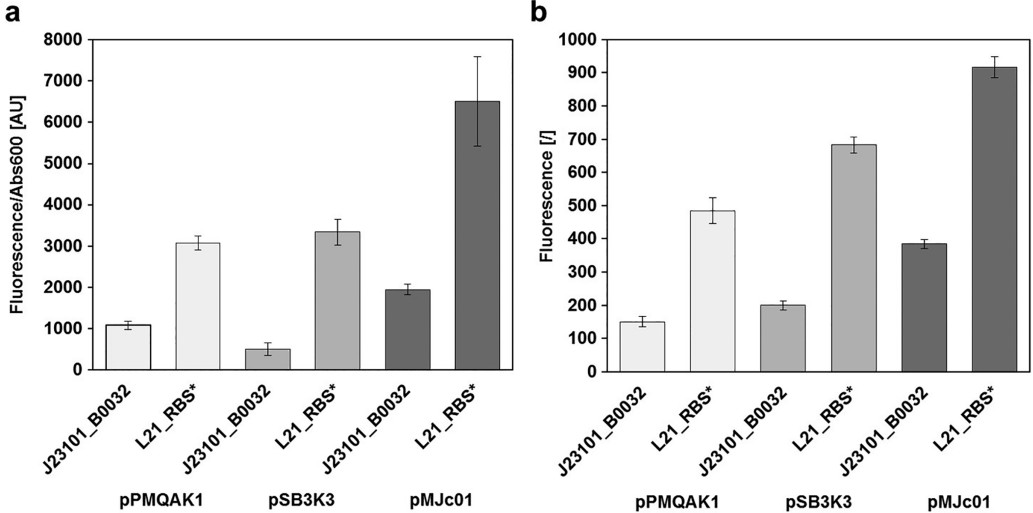

**Figure 3 Relative GFPmut3b reporter expression from different plasmid backbones.** GFPmut3b fluorescence intensity in *E. coli* transformed with pMJc01, pSB3K3, or pPMQA1 containing different regulatory sequences in the reporter expression device. Differences between mean values for triplicates of cells with expressed GFPmut3b and cells with empty vector ± standard deviation are shown. (A) GFPmut3b fluorescence intensity in cell cultures of *E. coli*. (B) GFPmut3b fluorescence intensity in cell lysates of *E. coli*.

with pPMQAK1. As judged from fluorescence of cell lysates, the expression ratios pPMQAK1:pSB3K3:pMJc01 were 1:1.3:2.5 for reporter constructs with J23101_B0032 regulatory sequence and 1:1.4:1.9 for reporter constructs with L21_RBS* regulatory sequence. Expression ratios based on cell culture fluorescence were 1:0.5:1.7 (for the reporter construct with J23101_B0032 regulatory sequence; the lower value for pSB3K3 fluorescence was attributed to measurement artefact due to greater differences in sample turbidity, see Supplemental Article S1) and 1:1.1:2.3 (for the reporter construct with L21_RBS* regulatory sequence), respectively.

Since we expected the higher expression levels with pMJc01 to be explained mainly by the higher plasmid copy numbers in *E. coli*, we decided to roughly determine the absolute average copy numbers of pMJc01 per *E. coli* cell. For this purpose, we used the quantification of restriction fragments after agarose gel electrophoresis. As a control and for comparison, we also included pSB3K3 in the analysis. Digestion of pMJc01 and pSB3K3 with *Not*I using two different amounts of isolated plasmids revealed the presence of approximately 18 copies for pSB3K3 and 26 copies for pMJc01. In addition, we verified these results with *Bsp*HI. restriction of isolated plasmids, resulting in 17 and 25 copies for pSB3K3 and pMJc01, respectively. We also attempted to include pPMQAK1 in the analysis, but the amount of plasmid isolated from the same number of cells as for the other two plasmids was always too low for quantification. Presumably, this was not due to lower copy number but rather to loss of the plasmid during alkaline lysis (see "Discussion").

Fluorescence measurements also show that the regulatory sequence L21_RBS* in *E. coli* is about 3-fold stronger than J23101_B0032 with reporter fluorescence ratios in cell lysates

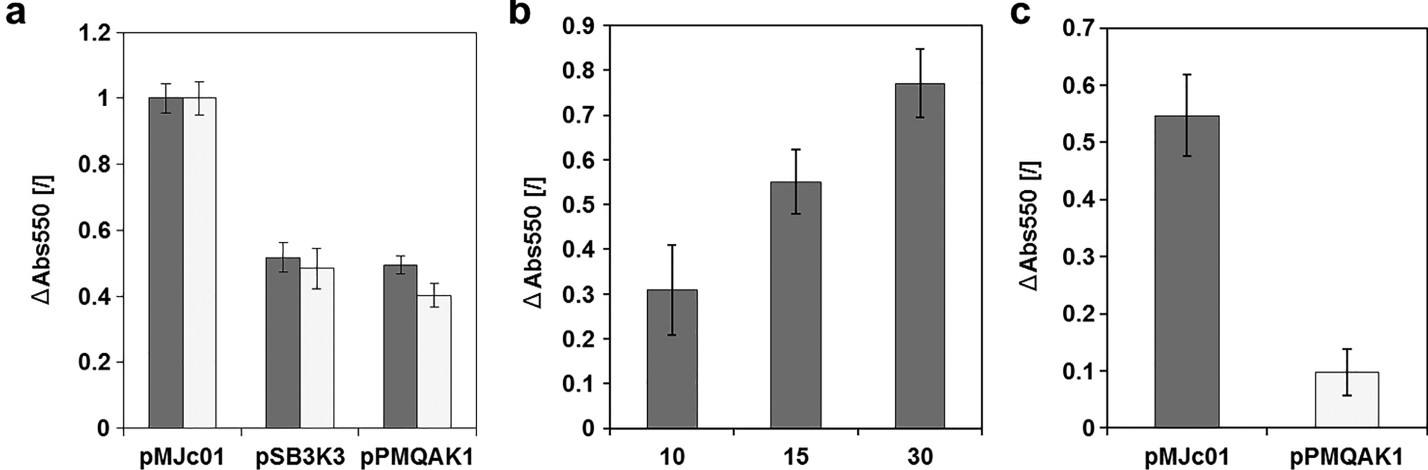

**Figure 4 Analysis of chicken cystatin as a reporter in *E. coli* and *Synechocystis* sp. PCC 6803.** Expression of chicken cystatin was determined with papain inhibition assay – high ΔAbs550 values indicate stronger inhibition and therefore higher quantities of cystatin in the tested lysates. Error bars represent ± standard deviation. (A) Relative plasmid copy number evaluation in *E. coli* with cystatin reporter. Light and dark gray bars represent two replications of the same experiment. The absolute values for both experiments were normalized (1 being the measurement with pMJc01) as the amount of cell material differed between repetitions. (B) Linearity of response for papain inhibition assay. 10, 15 or 30 μl of cell lysate (*E. coli* transformed with pMJc01 and cystatin expression device) were used. Samples were measured in duplicates. (C) Evaluation of cystatin reporter expression from plasmids pMJc01 and pPMQAK1 in *Synechocystis* sp. PCC 6803.

of 1:3.2, 1:3.4, and 1:2.4 when measured with pPMQAK1, pSB3K3, and pMJc01, respectively, and 1:2.8, 1:6.6, and 1:3.7 in cell cultures. The fluorescence determined with pSB3K3 in cultures differed significantly from the other values, which we attributed to a bias in the fluorescence measurement of samples with different turbidity, as mentioned above and shown in Supplemental Article S1.

## Plasmid characterization with cystatin as reporter gave comparable results to GFPmut3b in *E. coli*

To evaluate cystatin as a reporter for *E. coli* and cyanobacteria, a new expression construct with L21_RBS* regulatory sequence was prepared. The complete sequence of the pMJc01 vector with the reporter construct is deposited in GenBank under accession number MN201591. We used this construct to compare the relative copy numbers of pMJc01, pSB3K3, and pPMQAK1 in *E. coli* (Fig. 4A). Two separate experiments were performed, both resulting in ratios comparable to those obtained with GFPmut3b reporter (1:1.0:2.0 and 1.0:1.3:2.5 in the first and second experiments, respectively, with cystatin as reporter). In addition, we examined the linearity of the response of the papain inhibition assay used (Fig. 4B) and obtained results that were in good agreement with the expected ratio of the signals.

Since EV controls were also included in the experiment, we were able to evaluate the inhibitory effect of the chassis on the papain inhibition assay. The $A_{550}$ values of the samples containing lysates of cells with EV were only slightly lower compared to the control without the addition of cell lysate, therefore we concluded that there are no endogenous inhibitors in *E. coli* that could interfere with the cystatin reporter assay.
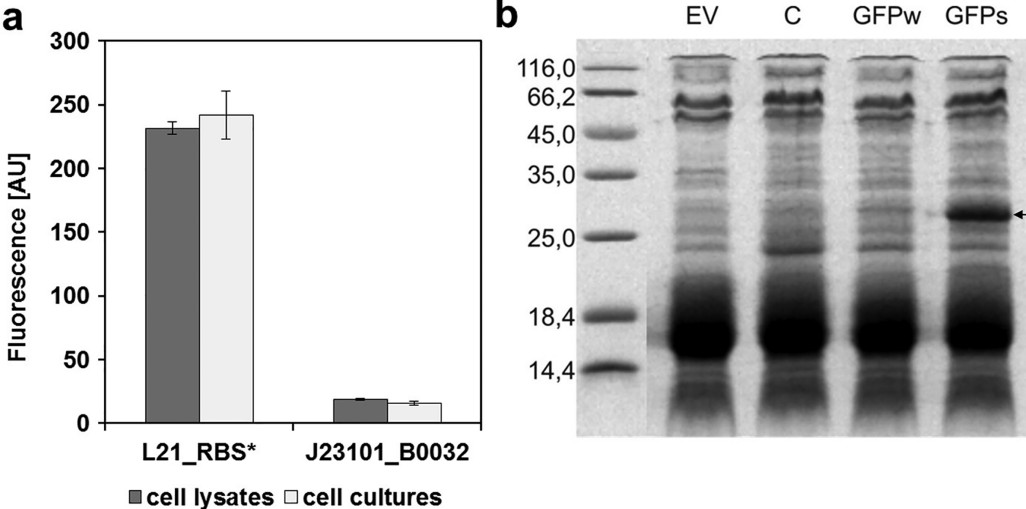

**Figure 5 Expression of GFPmut3b in *Synechocystis* sp. PCC 6803 transformed with pMJc01 vector.**
(A) GFPmut3b fluorescence of cell lysates and cultures of cyanobacteria *Synechocystis* sp. PCC 6803, transformed with pMJc01 vector with GFPmut3b expression under control of L21_RBS* or J23101_B0032. Error bars represent ± standard deviation. (B) SDS-PAGE of *Synechocystis* sp. PCC 6803 cell lysates; EV, cells transformed with empty pMJc01; C, cells transformed with pMJc01_L21_RBS*_*cystatin6803*; GFPw, cells transformed with pMJc01_BBa_I20260; GFPs, cells transformed with pMJc01_L21_RBS*_*GFPmut3b*. The protein band corresponding to *GFPmut3b* is marked with an arrow.

## pMJc01 is stable in *Synechocystis* sp. PCC 6803 and enables strong expression of GFPmut3b

To determine the replication ability of pMJc01 in *Synechocystis* sp. PCC 6803 at different growth stages, we used the reporter vector pMJc01_L21_RBS*_*GFPmut3b* (Fig. 2B). The fluorescence signal showed quite stable intensity values, which slightly decreased in the exponential growth stage. After 3 weeks of cultivation, no decrease in fluorescence was detected, demonstrating stable replication of pMJc01 in *Synechocystis* sp. PCC 6803 under antibiotic selection pressure. We also examined the amount of expressed GFPmut3b reporter in the stationary growth phase using SDS-PAGE and Coomassie Brilliant Blue staining and saw a distinct band of the appropriate size (26.9 kDa, Fig. 5B), indicating strong reporter expression under the regulatory control of L21_RBS*.

In *Synechocystis* sp. PCC 6803, the regulatory sequences analysed with pMJc01 showed similar reporter fluorescence values when measured in cultures or cell lysates (Fig. 5A). The relative activity of L21_RBS* compared to J23101_B0032 was even higher than in *E. coli*, with approximately 14-fold higher activity of L21_RBS* (12.4-fold when measured with cell lysates and 15.4-fold when measured with cultures).

## Cystatin is a suitable reporter protein for *Synechocystis* sp. PCC 6803 and showed approximately 5-fold stronger expression from pMJc01 compared to pPMQAK1

The cystatin expression construct was also analysed in *Synechocystis* sp. PCC 6803 (Fig. 4C) with vectors pMJc01 and pPMQAK1. We were able to implement the papain

inhibition assay for use with *Synechocystis* sp. PCC 6803 without significant modifications. We only had to optimise the cell lysis step, as this can be more challenging compared to *E. coli*. Several different approaches were tested (see Supplemental Article S1), but we found that sonication was best for this purpose. Although chemical methods also enabled effective cell lysis or facilitated sonication by decreasing the sonication times required, the chemicals used interfered with the inhibition assay as controls without cell lysate but with lysis buffer added also showed strong papain inhibition. The sonication time was optimised so that after the last sonication interval and centrifugation for 5 min at 14,000 g there was no green pellet left, indicating complete lysis. We again used EV controls to determine the background papain inhibition of the chassis and found that it was negligible, although slightly higher than in *E. coli*. The designed assay was used to compare pMJc01 and pPMQAK1 in S*ynechocystis* sp. PCC 6803. The amount of cystatin detected was approximately 5-fold higher in cells transformed with pMJc01 than in cells transformed with pPMQAK1 (Fig. 4C).

## DISCUSSION

Despite some disadvantages that cyanobacteria have compared to established laboratory enterobacteria in genetic engineering, significant progress has been made recently to overcome obstacles and pave the way for cyanobacteria to be considered the "new *E. coli*" (*Ruffing, Jensen & Strickland, 2016*). The main advantages of cyanobacteria, in particular their autotrophic nature and relative ease of genetic manipulation, make these microorganisms very attractive hosts for biotechnology.

We focused on three drawbacks that the majority of currently used episomal vectors and reporters for cyanobacteria have: low biosafety (due to the fact that shuttle vectors contain broad hostrange replicators and allow conjugative transfer), low copy number (mostly au pair with the number of genome copies in *Synechocystis* sp. PCC 6803), and relatively high fluorescent reporter signal backgrounds resulting from cellular components. To address these weaknesses, we designed, constructed, and tested the pMJc01 shuttle plasmid and cystatin as an alternative reporter.

The absence of mobilization genes is an essential requirement for improved vector biosafety, as it reduces the likelihood of horizontal gene transfer (*Wright, Stan & Ellis, 2013*). The majority of currently used broad host range replicative plasmids in cyanobacteria (*e.g.*, pPMQAK1) are based on the RSF1010 replicon, which contains mob regions that enable plasmid mobilization in the presence of an auxiliary vector. Our novel pMJc01 plasmid has an RSFmob I-based replicon that lacks the mobilization protein coding sequences as well as the origin of transfer (oriT) and has an undetectable mobilization frequency that is minimally 5 orders of magnitude lower than that of the RSF1010 replicon (*Katashkina et al., 2007*). Thus, the novel pMJc01 plasmid meets the standard of biosafety (*Wright, Stan & Ellis, 2013*). Although this simultaneously means that conjugation is no longer feasible as a means of plasmid transfer to cyanobacteria, we believe this is a small sacrifice, as both electroporation (*Chen et al., 2013*; *Tsujimoto et al., 2015*) and natural transformation (*Nies et al., 2020*) protocols have high efficiencies in many cyanobacterial species.

Omission of the RSF1010 mobilization regions further contributes to the superior properties of pMJc01 over RSF1010-based vectors by avoiding loss of the plasmid during isolation by alkaline lysis, a phenomenon that has been described for IncQ plasmids. Kok and coworkers (*Kok, Arnberg & Witholt, 1989*) demonstrated that nicking at *ori*T causes the formation of single-stranded circular DNA, visible on agarose gel after electrophoresis of plasmids isolated by alkaline lysis. A DNA species with high electrophoretic mobility was previously observed for the pPMQAK1 plasmid during our work and also by other researchers (P. Lindblad, personal communications, see Fig. S4). Relatively low copy number combined with loss of double-stranded plasmid DNA during alkaline lysis results in low isolation yield and complicates cloning in *E. coli* and transfer into cyanobacteria. Since the *mob* regions are absent in pMJc01, this drawback was not observed during our experimental work. We were always able to isolate decent amounts of pMJc01 from *E. coli* and had better electroporation efficiency results compared to pPMQAK1. However, we observed some structural instability of pMJc01 (see Supplemental Article S1), but only when it was propagated under conditions with decreased LacI activity. Based on the observed dependence of structural instability on LacI activity, we speculate that it may be the result of an unknown interaction with $P_{lacUV}$, which controls the expression of replication proteins in the RSFmob-I replicon. Unfortunately, we were not able to determine the nucleotide sequence of the elongated plasmid, which could help us find a mechanistic explanation for this interesting phenomenon. Whether the $P_{lacUV}$ is responsible for the observed phenomena could be determined by replacing it with other promoters (an option further discussed in Supplemental Article 1) and observing the stability of the plasmid in different genetic backgrounds. Testing other promoters is also an interesting option in case of unwanted crosstalk in experimental setups using lactose operon-based regulatory systems, as well as for further improvements of pMJc01, since another promoter could increase (or decrease) plasmid copy number.

The second goal for the pMJc01 plasmid was to achieve higher copy numbers in both *E. coli* and *Synechocystis* sp. PCC 6803. In *E. coli*, this would facilitate the cloning and construction of synthetic elements to be transferred into cyanobacteria. In *S.* sp. PCC 6803, a higher copy number would allow higher protein expression levels, which are usually difficult to achieve in cyanobacteria but are often desirable. Exceptions include the use of strong regulatory elements such as the "super-strong" $P_{cbcB}$ promoter (*Zhou et al., 2014*) and a combination of the $P_{psbA*}$ promoter with two RBS variants (*Wang et al., 2018*). Although development of stronger promoters has made an important contribution to the cyanobacterial toolbox, plasmids with higher copy numbers might provide more stable and reliable high expression profiles, because for regulatory elements (especially RBS), the specific genetic context can significantly alter their activity, resulting in different expression levels for different coding sequences (*Englund, Liang & Lindberg, 2016*). The use of promoters with lower absolute activities but other desirable properties, such as good inducibility and low leakiness, is also sometimes advantageous. Using pMJc01 in combination with L21_RBS*, we were able to achieve high expression levels of the GFPmut3b reporter in both chassis with visible protein bands in CBB-stained SDS-PAGE

gels. Especially in *Synechocystis* sp. PCC 6803 this is rarely achieved otherwise, neither with episomal nor with chromosomal expression.

The novel pMJc01 plasmid has a higher copy number than the RSF1010-based pPMQAK1 plasmid in both *E. coli* and *S.* sp. PCC 6803. Although copy numbers were determined by a relatively simple densitometric approach (*Projan, Carleton & Novick, 1983*), it is still regularly used (*e.g.*, *Valdelvira et al., 2021*). Thus, our results provide compelling evidence that pMJc01 has an approximately 2-fold higher copy number than pPMQAK1 in *E. coli*. This was determined using two different reporter systems (fluorescence-based GFPmut3b and cystatin activity) and two different regulatory sequences. These results were also confirmed by plasmid isolation and restriction fragment quantification. The obtained values were in agreement with the results of comparison of RSFmob-I and RSF10101 replicons (*Katashkina et al., 2007*), where they observed 2–3-fold higher copy number after 16 h of bacterial growth. Based on the estimated copy number of 10 for RSF1010-based plasmids (*Wang et al., 2012*), we suggest that pMJc01 has about 20–30 copies per chromosome in *E. coli* cells in the stationary growth phase, which we also confirmed with our restriction-based quantification. Higher copy number is also reflected in higher plasmid isolation yields. Based on fluorescence measurements at different growth stages, we concluded that pMJc01 copy number is approximately 2-fold higher in the stationary growth phase than in the exponential phase, a property previously observed for the RSFmob-I replicon (*Katashkina et al., 2007*).

Using cystatin as a reporter, we compared pMJc01 and pPMQAK1 in *Synechocystis* sp. PCC 6803 and showed approximately 5-fold higher reporter expression, which we attributed to the higher copy number of pMJc01. In addition, by GFPmut3b fluorescence measurements, we showed stable replication of pMJc01 in *S.* sp. PCC 6803 over the course of 3 weeks, with little change in copy number (Fig. 2B). We believe that the high copy number of pMJc01 in *S.* sp. PCC 6803 combined with a relatively strong regulatory sequence contributed to the high absolute levels of the GFPmut3b reporter, as shown by SDS-PAGE of cell lysates.

There are many RSF1010-based shuttle backbones that have been tested and used in cyanobacteria, *e.g.*, pPMQAK1 (*Huang et al., 2010*) and its derivatives (for example pJA2 (*Anfelt et al., 2013*)), RSF1010 replicon variants for constructing modular vectors (*Taton et al., 2014*), pVZ321 (*Zinchenko et al., 1999*) and its derivatives, such as pSL2680 (*Ungerer & Pakrasi, 2016*) and pSL1211 (*Ng et al., 2000*), and recently tested backbones from the SEVA (Standard European Vector Architecture) repository (*Ferreira et al., 2018*). These vectors have been used for a range of studies, from characterization of synthetic biology genetic elements (*Huang & Lindblad, 2013*; *Camsund, Heidorn & Lindblad, 2014*; *Badary et al., 2015*; *Englund, Liang & Lindberg, 2016*) to protein overexpression (*Anfelt et al., 2013*; *Ng et al., 2000*), including genome-modifying enzymes (*Ungerer & Pakrasi, 2016*). Despite the widespread use of RSF1010-based vectors, RSFmob-I-based plasmids, such as pMJc01, can be considered an ideal alternative for a variety of applications due to their stability during plasmid isolation, higher copy number in both *E. coli* and cyanobacteria, improved biosafety, and easier manipulation. Although some backbones devised from the native *Synechocystis* sp. PCC 6803 plasmids

(*e.g.*, *Armshaw et al., 2015*; *Jin et al., 2018*) also show higher copy numbers compared to RSF1010-based plasmids, they still lack the broad host range, another advantage of RSFmob-I-based plasmids.

Protein reporters are indispensable tools for the quantitative characterization of regulatory elements involved in the assembly of complex genetic circuits (*Decoene et al., 2018*). To this end, reporters should be functional in the chosen microbial strain and have a broad dynamic range covering common expression levels. Additionally, it is desirable that they allow for both high-throughput and single-cell analyses (*Martin, Che & Endy, 2009*). In cyanobacteria, fluorescent proteins are most commonly used as reporters, but they can have notable drawbacks because photosynthetic pigments in cyanobacteria can potentially interfere with the measured fluorescence signal by competing for the absorption of excitation light, with the re-absorption of emitted fluorescence, and by emitting intrinsic fluorescence that affects the excitation or emission of the recombinant fluorescent proteins. Therefore, the selection of fluorescent reporters with spectral properties that cause minimal interference with the cellular background is critical (*Ruffing, Jensen & Strickland, 2016*). The use of non-fluorescent reporters for cyanobacteria could therefore be beneficial and independent of the species or strain used. Moreover, it could also allow orthogonality when using light-inducible regulatory systems, which have already been successfully implemented in cyanobacteria (*Badary et al., 2015*) and whose activity could change with exposure to the excitation light used in reporter measurements.

To this end, we tested cystatin, a 13 kDa protein inhibitor of cysteine proteases that has been used previously as a qualitative reporter (*Škrlj, Erčulj & Dolinar, 2009*) but never as a quantitative reporter to characterise regulatory elements in vectors. Among several cystatins, we chose hen egg white cystatin, a stable and highly efficient inhibitor of papain and related proteases (Ki values in the pM range) (*Machleidt et al., 1989*). Cystatin expression was successful in both *E. coli* and *Synechocystis* sp. PCC 6803, and cystatin expression-based plasmid characterization in *E. coli* yielded comparable results to those obtained with GFPmut3b, demonstrating the potential of cystatin for quantitative characterization of regulatory genetic elements. The latter is further supported by the linear response of activity against the amount of cell lysate and thus against cystatin concentration. In *Synechocystis* sp. PCC 6803, we could directly compare cystatin production from pPMQAK1 and pMJc01 plasmids and detected significantly higher cystatin expression with pMJc01.

Although the expressed levels of cystatin appeared to be lower than those of GFPmut3b (the latter was detectable by SDS-PAGE and Coomassie Brilliant Blue staining in both organisms, whereas cystatin was not), it still allowed quantification by papain inhibition assay. This correlates with a known advantage of reporter systems based on enzymatic activity over fluorescent proteins—enzymes allow signal amplification, whereas with fluorescent probes one molecule provides one fluorophore that will photobleach with time, making detection of weakly expressed proteins difficult (*Martin, Che & Endy, 2009*).
The quantification range of cystatin could be further increased by using fluorescent papain substrates (*Škrlj, Erčulj & Dolinar, 2009*).

Cystatin lacks the main advantage of fluorescent reporters, namely *in vivo* real-time measurements. Quantification of cystatin requires cell lysis, which can be complicated by the presence of strong cell wall structures in some microorganisms, such as cyanobacteria. We had to test and adapt several protocols to achieve a satisfactory lysis compatible with the papain inhibition assay. On the other hand, we also found some biases in *in vivo* measurements with the GFPmut3b fluorescent reporter. Namely, we observed a strong inner filter effect when measuring fluorescence of *E. coli* cultures (see Supplemental Article S1), which causes erroneous normalization because the calculation of fluorescence per cell (the usual way of processing fluorescence measurements) can differ greatly when measuring at different optical densities. Although this can be solved by diluting all measured samples to the same optical density (as was done in our experiments), this is not usually the practice as it detracts from the simplicity of parallel in-plate measurements. Moreover, it is not practical for continuous *in vivo* measurements. In light of the above, we believe that although data acquisition is simpler with fluorescent proteins, making them valuable for rapid high-throughput screening, measurement bias, either due to the inner filter effect or other interactions with endogenous pigments, may be unacceptable for more accurate and absolute characterizations of biological parts. We believe that the use of an alternative reporter with a different type of signal generation should be used for verification of fluorescence measurements of biological parts of interest, and we propose cystatin as a promising reporter.

Another strong motive for using different reporters to characterize biological parts, especially regulatory sequences, is the strong sequence-specific influence of the downstream gene on their activity. For both promoters and RBS, it has been consistently shown that genetic context strongly influences expression levels (*Englund, Liang & Lindberg, 2016*). The use of different reporters has already been adopted for a detailed analysis of RBS in *Synechocystis* sp. PCC 6803 (*Thiel et al., 2018*).

## CONCLUSIONS

The evidence presented shows that the pMJc01 plasmid based on the RSFmob-I replicon has higher copy number in *E. coli* and *Synechocystis* sp. PCC 6803 and better biosafety and stability compared to the common RSF1010-based broad hostrange plasmid pPMQAK1. It can be used for easy cloning and biological part construction in *E. coli* as well as for protein expression in *Synechocystis* sp. PCC 6803. In both chassis, it should also prove useful for the characterization of genetic elements.

We also introduce a new non-fluorogenic reporter, cystatin, which is active and useful in both *E. coli* and *Synechocystis* sp. PCC 6803. It provides an alternative to fluorescent reporters in experimental setups where they are not applicable, as well as a useful tool for verifying the results of fluorescence-based measurements.

## ACKNOWLEDGEMENTS

We would like to thank Dr. Joanna Katashkina from Ajinomoto-Genetika Research Institute, Moscow, for her help in obtaining the RSFmob-I plasmid from the Russian National Collection of Industrial Microorganisms.

### Funding

The project was funded from university funds and by the Slovenian Research Agency (Structural Biology programme, award No. P1-0048). There was no additional external funding received for this study. The funders had no role in study design, data collection and analysis, decision to publish, or preparation of the manuscript.

### Grant Disclosures

The following grant information was disclosed by the authors:
Slovenian Research Agency: P1-0048.

### Competing Interests

The authors declare that they have no competing interests.

### Author Contributions

- Mojca Juteršek conceived and designed the experiments, performed the experiments, analyzed the data, prepared figures and/or tables, authored or reviewed drafts of the paper, and approved the final draft.
- Marko Dolinar conceived and designed the experiments, analyzed the data, authored or reviewed drafts of the paper, and approved the final draft.

### DNA Deposition

The following information was supplied regarding the deposition of DNA sequences:

The complete sequence of the pMJc01 vector with the reporter construct is available in GenBank: MN201591.

### Data Availability

The raw data is available in the Supplemental Files.

### Supplemental Information

Supplemental information for this article can be found online at http://dx.doi.org/10.7717/peerj.12199#supplemental-information.

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
