# Peer review of "A chimeric vector for dual use in cyanobacteria and Escherichia coli, tested with cystatin, a nonfluorescent reporter protein"

_PeerJ, doi:10.7717/peerj.12199_

## Round 0.1 · original submission · Major Revisions

Dear authors:

After carefully reading the reviewers' comments, I am obliged to request major revisions be done to the manuscript for me to further consider the manuscript for publication. All comments address the same issues, to state and demonstrate the expression of cystatin as a reporter, the characterization that the reporter is fully functional, and, more importantly, revise the English and grammar throughout the manuscript.

From reviewer 1, I kindly encourage you to assess all points, with an emphasis on points 1 and 3, also attend the addition of the references that are relevant to this manuscript.

From reviewer 2, there are numerous grammar issues highlighted by the reviewer that need attention. I find it critical to provide evidence that cystatin activity is present in the system, and I agree with the reviewer that an anti-His tag western blot is needed.

In line 448, I strongly suggest carrying a careful analysis to provide more insight into the role of P lacUV.

I find the manuscript interesting and provides further tools for researchers working on elucidating basic aspects of Synechocystis sp.

I strongly believe that addressing the concerns raised by the reviewers will improve the manuscript.

With best regards,

·

Basic reporting

no comment

Experimental design

no comment

Validity of the findings

no comment

Additional comments

no comment

Reviewer 2 ·

Basic reporting

no comment

Experimental design

no comment

Validity of the findings

no comment

Additional comments

The article describes the formation of a plasmid compatible with E. coli and in the cyanobacterium Synechocystis sp. PCC 6803, the plasmid pMJc01 can be introduced by electroporation into the bacterium Synechocystis sp. PCC 6803, and the plasmid carries the sequence coding for cystatin protein which is a papain inhibitor protein, which can serve as a reporter to evaluate the activity of regulatory elements in this type of bacteria. It is also postulated that it can be used as a tool for the expression of proteins in Synechocystis and others related-bacteria. However, I believe that it should be demonstrated more convincingly that cystatin is present in Synechocystis and that it has inhibitory activity.

147: next morning; replace for next day.

148: Why is it necessary to gradually increase the concentration of Kanamycin in the case of Synechocystis sp.?

179: L21_RBS*regulatory element. what does the asterisk mean?


207: What is LBK? please specify in the text.


257: Why in the papain inhibition assay, to test for the presence of the inhibitor (cystatin) was measured at 550 nm, in the original protocol was measured at 520 nm.



284: Indicate in Figure 1 the unique restriction sites (MCS if exist) that may be useful to exchange for others the promoter regions or gene of interest.


370 : In the Figure 4b. In the original activity assay the enzyme activity increases with increasing amount of enzyme used (ANALYTICAL BIOCHEMISTRY 47, 280-293 (1972)) and in the presence of inhibitor the activity decreases as the amount of inhibitor increases, as measured by Abs at 520 nm, as shown in reference 38. Why in the figure 4b the Abs increases as the amount of inhibitor (cystatin) increases, shouldn't it decrease?.


373: What do you mean by "chassis"? is the plasmid backbone?


386: mark with an arrow the GFP protein band in the figure 5b.


393: In the Figure 4C shows the relative copy number of the pMJc01 plasmid, which is higher than the pPMQAK1 plasmid, but this is not related to cystatin expression. The figure legends shows :"Evaluation of relative plasmid copy number in Synechocystis sp. PCC 6803 with cystatin reporter". It is essential to show convincingly the enzymatic activity of cystatin in Synechocystis, which is the main objective of the article. the authors point out that the cystatin gene carries a histidine-tag, so it would also be necessary to demonstrate its presence with an anti-histidine western blot.


408-409: The authors establish : "The amount of cystatin detected was approximately 5-fold higher in cells transformed with pMJc01 than in cells transformed with pPMQAK1 (Fig. 4c). but the figure shows that it is the relative amount of plasmid that is not related to cystatin expression.


448 : this is speculative "We attributed this to an unknown interaction with P lacUV , which controls the expression of replication proteins in the RSFmob-I replicon" please clarify.

---

## Round 0.2 · accepted · Accept

Dear authors:

After carefully assessing the comments by the reviewers and reading the latest version of the manuscript, I acknowledge that you have addressed all the comments and concerns raised by the reviewers. I thank you for providing a great paper that I am sure will get many citations.